# Understanding Prognostic Factors for Human Papillomavirus Vaccination: A Rural Community Case–Control Study

**DOI:** 10.3390/vaccines11101536

**Published:** 2023-09-28

**Authors:** Lara Colomé-Ceballos, Josep Lluís Clua-Espuny, Concepción Ceballos-García, Josep Clua-Queralt, Maria Jesús Pla-Farnós, José Fernández-Sáez

**Affiliations:** 1Sexual and Reproductive Attention, SAP Terres de l’Ebre, Catalonian Health Institute, 43500 Catalonia, Spain; 2Gynaecology Department, Hospital Verge de la Cinta de Tortosa, Catalonian Health Institute, 43500 Catalonia, Spain; 3EAP Tortosa Est. Primary Care, SAP Terres de l’Ebre, Catalonian Health Institute, 43500 Catalonia, Spain; jcluaq@gmail.com; 4Unitat de Suport a la Recerca Terres de l’Ebre, Foundation University Institute for Primary Health Care Research Jordi Gol i Gurina (IDIAPJGol), 43500 Tortosa, Spain; jfernandez@idiapjgol.info; 5Santa Creu de Jesús Hospital, Salut Tortosa, 43590 Catalonia, Spain; concepcion.ceballos@saluttortosa.cat; 6Gynaecology Department, Hospital Universitari de Bellvitge, Carrer de la Feixa Llarga, s/n, L’Hospitalet de Llobregat, 08907 Barcelona, Spain; mjpla@bellvitgehospital.cat; 7Faculty of Nursing, Terres de l’Ebre Campus, Rovira i Virgili University, 43500 Tortosa, Spain

**Keywords:** HPV vaccination, HPV infection, HPV vaccine acceptance, cervical cancer screening, sexually transmitted diseases

## Abstract

HPV vaccination coverage rates can vary depending on several factors. The main objective of this study is to identify possible independent prognostic factors that have an impact on HPV vaccination in a rural community, specifically related to sexual and reproductive health. A case–control, retrospective, community-based study was carried out on women aged 15 to 40 in the primary health centers of Southern Catalonia’s Terres de l’Ebre region, Spain, from 1 January 2020 to 31 December 2022. A random sample of 520 women with an average age of 29.3 (SD 7.8) years old was included in the study. Independent prognostic factors: age OR 0.680 (95% CI: 0.635–0.729, *p* < 0.001), immigrant origin OR 0.215 (95% CI: 0.109–0.422, *p* < 0.001), and HPV PCR OR 7.402 (95% CI: 2.504–21.880, *p* < 0.001). The variables that showed a barrier effect for HPV vaccination were age (OR 0.680, 95% CI 0.635–0.729, *p* < 0.001), and immigrant origin (OR 0.215, 95% CI 0.109–0.422, *p* < 0.001). The variable that showed a facilitating effect for HPV vaccination was HPV PCR (OR 7.402, 95% CI 2.504–21.880, *p* < 0.001).

## 1. Introduction

The human papillomavirus (HPV) vaccine is a critical component of the World Health Organization’s (WHO) 2030 Objectives program. This initiative places a strong emphasis on achieving high vaccination rates among women, with a specific target of attaining a 90% vaccination coverage rate among 15-year-old females [1]. The program aims to strengthen immunization efforts in primary healthcare [2]. However, its implementation shows substantial variability worldwide, with a global estimated coverage rate of only about 15% [3,4].

Regions undergoing demographic changes, such as the one in which this study was conducted, especially due to immigration phenomena, rurality, and socioeconomic characteristics [5,6,7,8,9], should not only evaluate the current status of the WHO program’s implementation but also explore ways to make its execution easier within the context of these demographic shifts. Additionally, they should address changes in the HPV guidelines and how to identify and address potential risk subgroups that may arise as a result [10,11,12,13].

On one hand, it is known that vaccination coverage rates vary due to numerous factors [14]: income level [3,4,15], accessibility [15,16,17], educational attainment [18,19], immigration status [20,21], smoking habits [22,23], and self-perception [24]. Moreover, most factors described are particularly evident in rural communities, resistant to cultural changes and exposed to deficits in basic resources such as the supply, monitoring, and recording of vaccinations of public interest and associated with lower vaccination coverage compared to urban regions. Understanding of these specific factors may help to devise targeted strategies for enhancing HPV vaccination coverage in rural locale communities [25,26,27,28].

On other hand, facilitating factors have been described for vaccination such as practicing safe sex and undergoing regular sexual and reproductive health screenings [29]. Likewise, women who have had Pap smears within the past year are more likely to have received the HPV vaccine [30]; and adolescent females who have received the HPV vaccine are more likely to undergo health examinations, including tests for sexually transmitted infections [31]. In addition, women who have received the HPV vaccine are more inclined to practice safer sexual behaviors when compared to their unvaccinated counterparts [32]. The recommended population screening for women aged 30–35 and older prioritizes HPV detection as the preferred test over conventional cytology. However, since this age group coincides with lower rates of HPV vaccination, it may be faced with new barriers to follow-up vaccination. The recent inclusion of boys in systematic vaccination programs [33] is expected to have a pivotal role in altering the landscape of HPV vaccination [34].

Effectively addressing these challenges necessitates a multifaceted approach that acknowledges the wider context of rural communities, takes into account their unique obstacles, and caters to their specific needs. The primary objective of this study is to ascertain potential autonomous prognostic determinants that impact HPV vaccination within a rural community, with a specific focus on aspects related to sexual and reproductive health.

## 2. Materials and Methods

### 2.1. Study Design

A retrospective, case–control, community-based study was carried out on women aged 15 to 40 in the primary health centers of Southern Catalonia’s Terres de l’Ebre region, Spain, with data available up to 31 December 2022.

### 2.2. Study Scope

The Terres de l’Ebre region (Appendix A) comprises four districts, 52 municipalities, and has a total population of 191,791 inhabitants. The population density is 54.5 inhabitants per square kilometer, which is significantly lower than the average population density in Catalonia. This region is also recognized as part of the demographic phenomenon referred to as “emptied Spain” [5,6,7]. The region demonstrates a progressive aging trend due to a negative migratory balance and a low birth rate, resulting in a higher aging index (162.7) compared to Catalonia (127.1) and Spain (118.43) [8,9,35]. Additionally, the average income per inhabitant is 77.4%, which is lower than the overall average for Catalonia (100%) [10]. A previous article [11] has already described the study population and provided information about their demographics, age at vaccination, vaccination rate, socioeconomic status, and other relevant factors.

### 2.3. Patients/Subjects of Study

The study’s target population encompassed 24,415 women aged 15 to 40 residing in the Terres de l’Ebre region, constituting 94.7% of the total number of women in that age group registered within the territory.

For this case–control study, a randomized sample size of 520 participants between the ages of 15 and 40 years, who are registered in the territory and have an active medical history in any of the healthcare centers within it, was calculated. The study aimed to include 182 cases of vaccinated patients and 338 controls of unvaccinated patients. The calculations were based on the following parameters: alpha risk of 0.05, beta risk 0.15, and a bilateral contrast to detect a minimum odds ratio of 0.5. A rate of 0.3 exposed in the control group is assumed, and a loss of 10% in follow-up has been estimated. The POISSON approach has been used (https://www.imim.es/ofertadeserveis/software-public/granmo/, accessed on 23 January 2023). The STROBE checklist was used for the review of items included.

### 2.4. Observation Period

All the data pertain to the clinical history of the subjects up to 31 December 2022.

### 2.5. Inclusion Criteria

The inclusion criteria were as follows:−Availability of clinical record at any of the health centers in the territory (HC3).−Women who have been residents for at least five years in one of the four regions of the Terres de l’Ebre territory (Baix Ebre, Montsià, Terra Alta, and Ribera d’Ebre), regardless of their place of birth.

### 2.6. Exclusion Criteria

The exclusion criteria were as follows:−Patients without active or open clinical history and/or with insufficient data or inaccessibility to said data.−Disease with vital prognosis of less than one year.

### 2.7. Variables

#### 2.7.1. Dependent Variable: HPV Vaccination Status

The vaccination status was classified as “correctly vaccinated” if the individual received either 2 doses before their 15th birthday or 2 or 3 doses after their 15th birthday, as well as for immunocompromised patients [36,37,38,39]. The patients were classified as non-vaccinated if they had never received a dose of HPV vaccination and incompletely vaccinated if they had received one dose.

The participants in this study were categorized into two groups based on their vaccination status, as per the criteria outlined by the World Health Organization (WHO) and the Centers for Disease Control and Prevention (CDC) [36,37]. Since 2017, all individuals who received vaccines subsidized by the public health system in the study region were administered the nonavalent vaccine (9vHPV), irrespective of the type of access to the vaccine. Patients under 15 years of age receive two doses, while those of 15 years or more receive three doses. Before 2017, the financed vaccine was the quadrivalent (4vHPV), and all patients received three doses until 2014, two doses from 2014 to 2017 if they were less than 15 years of age, and three doses in the rest of the patients. In the case of the patients vaccinated after medical counselling, the health personnel decide what type of vaccine is administered to the patient [40,41]. The number of doses administered, the type of vaccine, and the date of administration were included in the study.

#### 2.7.2. Independent Variables

-Demographic variables: age, ethnic origin, immigrant origin, and educational level.

Age was divided into intervals in Table 1 as a categorical variable with ordinal properties, as a binary categorical variable in Table 3, and as a continuous variable in the rest of the calculations. Immigrant origin was defined as individuals who were born in a different country but have moved to Spain to live and work. Ethnic origin was defined based on the country of origin as well as race, and educational level is defined as illiterate (inability to read or write), mandatory primary/secondary schooling, or higher education (beyond secondary high school level, such as bachelor’s, master’s, and doctoral degrees). These data were recorded in the patients’ clinical history.
−Variables related to cervical cancer screening: first registered cervical cytology, result of the first registered cervical cytology, first registered cervical PCR HPV test, and result of the first registered HPV PCR test.−Variables related to sexually transmitted diseases: first registered PCR sexually transmitted disease (STD) test in the vagina or cervix, result of the first registered PCR STD test in the vagina or cervix, and history of any diagnosis of an STD or HIV in the medical record. The STDs that are included in the test available at our laboratory are: *Chlamydia trachomatis*, *Neisseria gonorrhoeae*, *Trichomonas vaginalis*, *Ureaplasma* spp., and *Mycoplasma* spp.−The variables related to sexual and reproductive health included factors such as childbirth, breastfeeding, voluntary termination of pregnancy or abortion, contraception methods, and the use of condoms.−Other clinical variables: smoking and immune status.

The variables have been categorized based on the individual’s vaccination status, captured at the time of HPV vaccination, and the results have stratified into different age groups or periods.

### 2.8. Data Collection and Information Sources

All study participants belong to the Catalan Institute of Health (ICS), which manages the area through 11 Primary Care Centers (EAPs). Data were obtained retrospectively through an anonymized database provided by the Service of Technology and Communication. The data are recorded using the International Classification of Diseases (ICD-10) and sent to the principal investigator in an anonymized format. Data were supervised and analyzed according to the General Data Protection Regulation of Spain/Europe as of 1 February 2017. The study was conducted in accordance with the most relevant standards on data handling, experimental context with patients, ethics, and data protection and privacy, following Directive 95/46/EC (protection of individuals with regard to the processing of personal data and on the free movement of such data). All collected data were compiled and stored in a dedicated repository specifically created for this study. The repository was then handed over to the principal investigator for analysis. Prior to conducting the study, the research protocol underwent thorough ethical evaluation and received approval from the Ethics Committee of the Jordi Gol Primary Care Research Institute, with the registration number 21/064-P.

The datasets utilized for this project include:The Health Plan of the Terres de l’Ebre Healthcare Region 2021–2025 [42]: This strategic document outlines the goals, priorities, and actions that will guide healthcare services in the Terres de l’Ebre region of Catalonia, Spain, from 2021 to 2025.The HC3 Patient Episode Dataset for Catalonia (CatSalut, Department of Health): This dataset contains demographic and clinical information on inpatient and outpatient care in Catalan hospitals. It provides details on patient episodes of care, including hospital admissions, outpatient visits, and diagnostic tests.The shared clinical information database of the 11 primary care teams managed by the Catalan Health Institute: This database includes clinical data, symptoms, tests, diagnoses, comorbidities, prescribed medication, referrals to secondary and tertiary care, and the alive/dead status of 97.7% of the residents in the territory.Pharmacological variables collected from the Integrated System of Electronic Prescription (SIRE): This dataset captures information on prescribed medications.Data provided by the Statistics Institute of Catalonia: This dataset includes information on gross household income per capita relative to the Catalonia average (100%), population density per square kilometer, and aging index compared to Catalonia (100%) [7,8,10].

Data on these factors was automatically collected whenever possible or manually collected otherwise.

### 2.9. Statistical Analysis

The vaccination coverage rate was calculated by dividing the number of patients with complete vaccination by the total target population. Data are presented using frequencies and percentages for categorical variables and using means and standard deviations for continuous variables, according to region, vaccination status, and age. To detect differences between the two groups, the chi-square test for categorical variables and the Mann–Whitney U test for continuous variables were used.

The study employed binary logistic regression using the backward stepwise method of Wald to identify the independent predictive variables associated with HPV vaccination. The backward stepwise method of Wald was used to eliminate variables that did not significantly contribute to the model while identifying those that were significantly associated with vaccination. The results were presented in the form of odds ratios (ORs) and their 95% confidence intervals. The ORs enabled the determination of the probability of vaccination for each predictive variable while considering the effect of other variables in the model. The variables analyzed included age, smoking, ethnic origin, immigrant origin, offspring, breastfeeding, contraception, cervical cytology, HPV PCR test, and STD PCR test. The association between variables was calculated using the OR, with a statistical significance level of *p* < 0.05 established, and 95% CI. The association between variables in the logistic regression model was evaluated using the area under the curve (AUC) of the receiver operating characteristic (ROC) curve. By utilizing the kappa concordance methodology, researchers assessed the agreement between the outcomes derived from the multivariate regression analysis and the real vaccination status against HPV. Specificity was employed to determine the proportion of true negative results (non-vaccinated individuals correctly identified as non-vaccinated) among all individuals who were not vaccinated, while sensitivity measured the proportion of true positive results (vaccinated individuals correctly identified as vaccinated) among all individuals who were vaccinated. All analyses were performed using the IBM SPSS Statistics 20 statistical package.

## 3. Results

A randomized sample of 520 cases was included, with an average age of 29.3 (SD 7.8 years old). HPV vaccination displayed a significant association with various factors, including regular Pap smear screenings (*p* < 0.001), contraception usage (*p* = 0.008), parenthood status (*p* < 0.001), and the practice of breastfeeding (*p* < 0.001). In the logistic regression analysis, the age (OR 0.680, 95% CI 0.635–0.729, *p* < 0.001), immigrant origin (OR 0.215, 95% CI 0.109–0.422, *p* < 0.001), and HPV PCR (OR 7.402, 95% CI 2.504–21.880, *p* < 0.001) emerged as independent facilitators of HPV vaccination.

### 3.1. Basal Characteristics

Statistically significant differences between vaccinated vs. unvaccinated groups are shown for average age in Table 1 and race, immigrant origin, and educational level in Table 2. In total, 27.3% (95% CI 23.3–31.2) of unvaccinated women have an immigrant background and differences by ethnic origins (*p* = 0.044) and immigrant status (*p* < 0.001) are shown in relation to vaccination status.

A total of 175 (33.6%) individuals were classified as correctly vaccinated. Among the vaccinated patients, the average number of doses administered was 2.54 (SD 0.59). Moreover, the probability of successfully finishing the entire vaccination schedule once initiated exhibited a 25-fold increase (95% CI 11.74–53.21; *p* < 0.001).

Significant statistical differences (Table 3) were observed between the screening tests performed and vaccination status, as well as the results of the cervical cancer screening tests and the mean ages of the different groups (*p* < 0.001), except for the mean age in the different results of the HPV PCR test (*p* = 0.125).

The average age of individuals without a recorded STD PCR test was 29.4 years old (SD 7.8), while the average age of those with at least one test recorded was 28.9 years old (SD 7.5) (*p* = 0.793). No statistically significant differences were found in the mean age of individuals who had undergone an STD PCR test or not or in the various outcomes of the test (Table 4).

Significant disparities (Table 5) were observed between the vaccinated and unvaccinated groups across several variables: mean age (21.8 (SD 5.2) vs. 33.1 (SD 5.9), *p* < 0.001); use of contraception (*p* = 0.008); parenthood (*p* < 0.001); breastfeeding practice (*p* < 0.001); and regular Pap smear screenings (*p* < 0.001).

Related to other variables, no significant differences were found in terms of immune status alterations (*p* = 0.170) as shown in Appendix B. In relation to tobacco use, 15.0% (95% CI 11.83–18.16) of the participants were registered as smokers with a statistically significant difference (*p* < 0.001) as shown in Appendix C. Non-vaccinated smoking patients were significantly older than non-vaccinated non-smoking patients (35.5 (SD 4.3) vs. 33.0 (SD 5.7), *p* = 0.002).

### 3.2. Prognostic Factors of HPV Vaccination

The initial model included the following variables: age, cervical cytology test, HPV PCR test, STD PCR test, offspring, breastfeeding, contraception, smoking, immune status, ethnic origin, and immigrant origin. The results (Table 6) showed age and immigrant origin as significant barriers to HPV vaccination. Conversely, cytology and HPV PCR are facilitators for vaccination.

Regarding the ROC curve analysis (Figure 1), the variable age exhibited an area under the curve (AUC) of 0.917 (95% CI 0.89–0.94), indicating strong discriminatory power; immigrant origin, 0.575 (95% CI 0.52–0.62); cervical cytology performed, 0.633 (95% CI 0.52–0.62); and HPV PCR performed, 0.498 (95% CI: 0.44–0.55).

The kappa concordance methodology showed good results, with a Kappa index of 0.755, sensibility of 0.81 (95% CI 0.75–0.86), specificity of 0.93 (95% CI 0.91–0.96), positive predictive value of 0.87 (95% CI 0.82–0.92), and negative predictive value of 0.90 (95% CI 0.87–0.93). Figure 2 shows the probability distribution of the forecast from the logistic regression model.

Eventually, three target risk groups were identified related to challenges in accessing HPV vaccination: women over 25 years old, women of immigrant origin, and women without regular cervical cancer screening, and the probability of successfully finishing the entire vaccination schedule, once initiated, exhibited a 25-fold increase (95% CI 11.74–53.21; *p* < 0.001). These findings underscore the significance of considering age, immigrant status, and accessibility to cervical cancer screening when addressing the issue of HPV vaccination access.

## 4. Discussion

A randomized sample of 520 cases was included, with an average age of 29.3 (SD 7.8) years. HPV vaccination displayed a significant association with various factors, including regular Pap smear screenings (*p* < 0.001), contraception usage (*p* = 0.008), parenthood status (*p* < 0.001), and the practice of breastfeeding (*p* < 0.001). In the logistic regression analysis, the age (OR 0.680, 95% CI 0.635–0.729, *p* < 0.001), immigrant origin (OR 0.215, 95% CI 0.109–0.422, *p* < 0.001), and HPV PCR (OR 7.402, 95% CI 2.504–21.880, *p* < 0.001) emerged as independent facilitators of HPV vaccination.

The characteristics of rurality, low population density, demographic aging, and below-average income levels in the country [5,6,7,8,9,10,35] may be negatively linked to the administration of HPV vaccine doses [43,44,45]. However, our results differ from significant associations with Hispanic ethnicity, county poverty, household characteristics, and uninsured rates, which could be attributed to variations in geographic and time frames [46,47,48].

The results regarding age can be explained by the school calendar factor. In Spain, the introduction of HPV vaccines into vaccination schedules began in 2008. Consequently, only individuals born after 1996 were integrated into school vaccination programs, enabling them to receive publicly funded HPV vaccination while attending school [49]. This creates a distinction from patients born before 1995. If these patients choose to get vaccinated, they require medical advice and must purchase the vaccine using their own funds. Altogether, these factors have become barriers to accessibility for HPV vaccination for immigrant women [17,18,19,20,21]. The study area has a higher percentage of immigrant population compared to the averages documented for Catalonia [50]. Lack of knowledge about the vaccination schedule also poses a barrier for them to access the catch-up option for funded vaccination. These results are similar to those of other studies [17,18,19] that illustrate noteworthy disparities in the initiation of HPV vaccination, influenced by factors such as ethnic background [51,52,53,54,55], parental attitudes [56], limited awareness [57,58], and financial constraints [59].

The findings related to cervical cytology or Pap smear closely paralleled established evidence [11] while also shedding light on certain distinctions, such as a reduced screening coverage and an elevated proportion of abnormal results in contrast to Spain’s national data [60]. Comparable patterns were evident in HPV PCR testing, with the younger vaccinated cohorts displaying elevated rates of positive infections. Moreover, given that PCR testing was identified as a contributing factor to vaccination, ironically, healthcare professionals should view the infection as an opportunity to increase vaccine coverage rates (VCRs) and emphasize the accessibility of healthcare professionals in this regard. These findings provide additional validation for the evidence-based suggestion to implement HPV PCR screening across the general population, as endorsed by scientific associations [61]. In 2019, the Ministry of Health, Consumption, and Welfare modified the Royal Decree on common services of the National Health System, through which the autonomous communities were gradually required to replace cytological screening with HPV testing and opportunistic screening with population-based screening [13].

Individuals who engage in risky sexual behavior, particularly in relation to unwanted pregnancies, are also at a heightened risk of contracting STDs and HPV [62]. Furthermore, there is an observed correlation between cervical dysplasia and sexually transmitted infections, particularly among younger patients [63]. The results suggest that there may be an underdiagnosis of these diseases. Individuals who face barriers to accessing healthcare services may encounter difficulties in obtaining screening for sexually transmitted diseases and cervical cancer. Consequently, they may miss out on vital primary prevention measures such as vaccination, as well as important secondary prevention measures such as regular screening for cervical cancer.

In order to improve protection against HPV-related diseases, it is recommended to complete the full vaccination schedule, which typically consists of two or three doses. However, the study highlights that receiving the first dose of vaccination increases the likelihood of completing the full schedule by 25 times. Studies [64,65,66] have demonstrated that individuals who initiate the HPV vaccination series are more likely to complete the entire schedule if they receive a recommendation from their healthcare provider and have a reminder system in place. Furthermore, these coverage results still have room for improvement by reducing the number of doses, which appears to enhance adherence to the regimen [67]. The WHO has published a document that enables better access to vaccination by reducing vaccine doses. This way, it could be used as a single-dose vaccine in individuals aged 9 to 20 years [68].

Regarding the remaining variables, “offspring” and “breastfeeding,” no previous evidence has been found. Further research is needed on this topic. As for immune status, modifications in the screening interval should be considered based on the woman’s medical history and her risk factors [69], due to their heightened vulnerability to HPV infection [41]. The role of smoking is debated in numerous studies [22,23,24], but our study did not find smoking to be a prognostic factor for HPV vaccination.

There are some limitations, such as the lack of data on the male population and potential limitations due to a registration bias in some variables in addition to the data sources selected. In Spain, it was in the year 2008 that HPV vaccines began to be introduced into the vaccination schedules of the autonomous communities, and it was in the year 2023 that vaccination was included for 12-year-old male adolescents. The results suggest a possible relationship between HPV vaccination and sexual and reproductive health behaviors but do not imply causation. Additional research would be necessary to determine how changes in vaccination coverage correlate with accessibility to sexual and reproductive health services and their potential interplay with the age and gender of immigrant individuals, especially among newcomers and due to the new indication of the HPV vaccine for males.

## 5. Conclusions

The study identified as barriers age (OR 0.68, 95% CI 0.63–0.72, *p* < 0.001) and immigrant origin (OR 0.21, 95% CI 0.10–0.42, *p* < 0.001), which were found to decrease the likelihood of HPV vaccination, and HPV PCR (OR 7.40, 95% CI 2.50–21.88, *p* < 0.001) as a facilitator for HPV vaccination. Also, the accessibility to the initial vaccine greatly enhances the likelihood of completing the full vaccination schedule, increasing it by a remarkable 25 times. These findings highlight opportunities to improve HPV vaccination rates.

## Figures and Tables

**Figure 1 vaccines-11-01536-f001:**
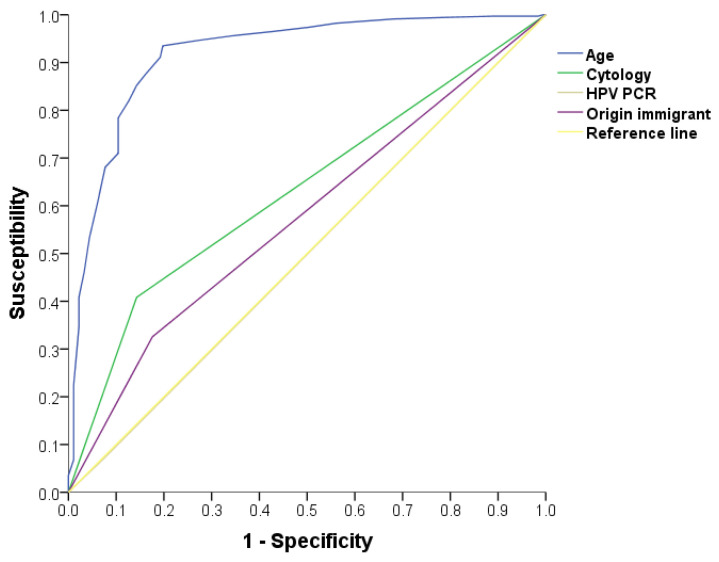
ROC curve of the logistic regression statistical model.

**Figure 2 vaccines-11-01536-f002:**
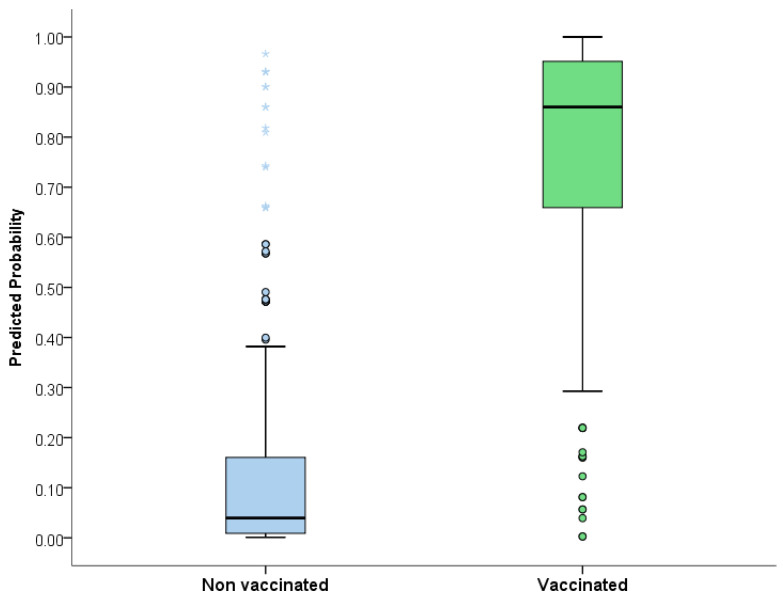
Probability distribution of the forecast from the logistic regression model.

**Table 1 vaccines-11-01536-t001:** Vaccination coverage rate by age groups.

Age Groups	Non-Vaccinated		Vaccinated		All		*p*
	*n*	%	*n*	%	*n*	%	
15–19	6	7.0	80	93.0	86	16.5	<0.001
20–24	16	19.5	66	80.5	82	15.8	
25–29	51	75.0	17	25.0	68	13.1	
30–34	109	89.3	13	10.7	122	23.5	
>35	156	96.3	6	3.7	162	31.2	
All	338	65.0	182	35.0	520	100.0	
Average age (SD ^1^) ^2^	33.1 (5.9)		21.8 (5.2)		29.3 (7.8)		<0.001

^1^ Standard deviation; ^2^ in years old.

**Table 2 vaccines-11-01536-t002:** Demographic variables by vaccination status.

	**Non-Vaccinated**	**Vaccinated**	**All**	*p* = 0.044
**Ethnic origin**	*n* (%)	*n* (%)	*n* (%)	
European	277 (62.7)	165 (37.3)	442 (85.0)	
African	35 (83.3)	7 (16.7)	42 (8.1)	
American	19 (73.1)	7 (26.9)	26 (5.0)	
Asian	7 (70.0)	3 (30.0)	10 (1.9)	
	Non-Vaccinated	Vaccinated	All	*p* < 0.001
**Immigrant origin**	*n* (%)	*n* (%)	*n* (%)	
Non-Immigrant origin	228 (43.8)	150 (28.8)	378 (72.7)	
Immigrant origin	110 (21.1)	32 (6.1)	142 (27.3)	
	Non-Vaccinated	Vaccinated	All	*p* = 0.081
**Education**	*n* (%)	*n* (%)	*n* (%)	
No education	21 (91.3)	2 (8.7)	23 (4.4)	
Compulsory education	90 (73.2)	33 (26.8)	123 (23.7)	
Higher education	66 (82.5)	14 (17.5)	80 (15.4)	

**Table 3 vaccines-11-01536-t003:** Cervical cancer screening test by vaccination status.

**Cervical Cytology Test**	**Non-Vaccinated**	**Vaccinated**	**All**	*p* < 0.001
	*n* (%)	*n* (%)	*n* (%)	
Performed	140 (41.4)	28 (15.3)	168 (32.3)	
Not Performed	198 (58.6)	154 (84.6)	352 (67.7)	
**Results of Cervical Cytology Test**	Non-vaccinated	Vaccinated	All	*p* < 0.001
	*n* (%)	*n* (%)	*n* (%) *	
Negative	136 (40.2)	19 (10.4)	155 (92.3)	
Positive	4 (1.2)	9 (4.9)	13 (7.7)	
**HPV ^1^ PCR ^2^ test**	Non-vaccinated	Vaccinated	All	*p* = 0.885
	*n* (%)	*n* (%)	*n* (%)	
Performed	23 (6.8)	13 (7.1)	36 (6.9)	
Not Performed	315 (93.2)	169 (92.9)	484 (93.1)	
**HPV ^1^ PCR ^2^ test**	Non-vaccinated	Vaccinated	All	*p* = 0.047
	*n* (%)	*n* (%)	*n* (%) *	
Negative	21 (87.5)	7 (58.3)	28 (77.8)	
Positive	3 (12.5)	5 (41.7)	8 (22.2)	

* From the total of those performed; ^1^ human papillomavirus; ^2^ polymerase chain reaction.

**Table 4 vaccines-11-01536-t004:** Sexual transmitted disease screening test by vaccination status.

**STD ^1^ PCR ^2^ test**	**Non-Vaccinated**	**Vaccinated**	**All**	*p* = 0.760
	*n* (%)	*n* (%)	*n* (%)	
Performed	20 (5.9)	12 (6.6)	32 (6.2)	
Not Performed	318 (94.1)	170 (93.4)	488 (93.8)	
**Results of STD ^1^ PCR ^2^ test**	Non-vaccinated	Vaccinated	All	*p* = 0.555
	*n* (%)	*n* (%)	*n* (%) *	
Negative	16 (76.2)	8 (66.7)	24 (72.7)	
Positive	5 (23.8)	4 (33.3)	9 (27.3)	

^1^ Sexually transmitted disease; ^2^ polymerase chain reaction; * from the total of those performed.

**Table 5 vaccines-11-01536-t005:** Variables of sexual and reproductive health by vaccination status.

**Offspring**	**Non-Vaccinated**	**Vaccinated**	**All**	*p* < 0.001
	*n* (%)	*n* (%)	*n* (%)	
No Offspring	166 (50.3)	164 (49.7)	330 (63.4)	
Offspring	171 (90.5)	19 (9.5)	189 (36.3)	
**Breastfeeding**	Non-Vaccinated	Vaccinated	All	*p* < 0.001
	*n* (%)	*n* (%)	*n* (%)	
No Breastfeeding	227 (56.8)	173 (43.2)	400 (76.9)	
Breastfeeding	111 (92.5)	9 (7.5)	120 (23.1)	
**Abortion**	Non-Vaccinated	Vaccinated	All	*p* = 0.729
	*n* (%)	*n* (%)	*n* (%)	
No Abortion	326 (65.1)	175 (34.9)	502 (96.5)	
Abortion	11 (61.1)	7 (38.9)	18 (3.5)	
**Contraception**	Non-Vaccinated	Vaccinated	All	*p* = 0.008
	*n* (%)	*n* (%)	*n* (%)	
No Contraception	40 (80.0)	10 (20.0)	50 (9.6)	
Contraception	76 (58.9)	53 (41.1)	129 (24.8)	
**Condom Use**	Non-Vaccinated	Vaccinated	All	*p* = 0.110
	*n* (%)	*n* (%)	*n* (%)	
No Condom Use	136 (72.3)	52 (27.7)	188 (36.1)	
Condom Use	50 (62.5)	30 (37.5)	80 (15.4)	

**Table 6 vaccines-11-01536-t006:** Logistic regression analysis.

Variable	*p*	OR	95% CI	
			Lower Limit	Upper Limit
Age	<0.001	0.674	0.625	0.726
Cervical cytology test performed	0.063	2.315	0.955	5.613
HPV PCR performed	0.001	7.337	2.352	22.888
STD PCR performed	0.337	0.569	0.180	1.797
Offspring	0.516	1.420	0.493	4.086
Breastfeeding	0.243	0.477	0.138	1.653
Smoking	0.462	1.398	0.572	3.415
Contraception	0.446	0.603	0.165	2.210
Immune status	0.142	3.966	0.629	25.005
Ethnic origin	0.388	0.523	0.120	2.280
Immigrant origin	0.011	0.298	0.118	0.754
**Results statistical model of logistic regression**				
Age	<0.001	0.680	0.635	0.729
Cervical cytology test performed	0.078	2.135	0.917	4.970
HPV PCR performed	<0.001	7.402	2.504	21.880
Immigrant origin	<0.001	0.215	0.109	0.422

## Data Availability

The data that support the findings of this study are available from the corresponding author (L.C.-C.) upon reasonable request.

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
