# Peer review of "Understanding Prognostic Factors for Human Papillomavirus Vaccination: A Rural Community Case–Control Study"

_vaccines, 2023, doi:10.3390/vaccines11101536_

Round 1

Reviewer 1 Report

Estimated Authors,

I've read with great interest your original article entitled "Understanding prognostic factors for HPV vaccination: a rural community case-control study". Through a random sampling from a very large population of over 20,000 female from the Catalonian region of Ebro, Colomé-Ceballos et al were able to identify the following predictive variable with regard to HPV vaccination: age COR 0.680, 95% CI 0.635-0.729, p <0.001), immigrant origin (OR 0.215, 95% CI 0.109-0.422, p <0.001), and HPV PCR (OR 7.402, 95% CI 2.504-21.880, p <0.001).

Albeit not particularly innovative, this article may be interesting for people involved in vaccination campaigns, particularly in areas characterized by progressive "medical desertification". However, I think that before its final acceptance, the present study would require substantial adjustements, and more precisely:

1) The article encompasses a lot of tables, and a large share of that could be collapsed by shifting to column variables vaccinated/not vaccinated that are now reported as a row; this approach would allow a more appropriate appraisal by the readers.

2) I've noticed that the average number of delivered doses was well below the unity. That means that a substantial share of subjects had an incomplete vaccination status rather a non-vaccinated status. Therefore, it would be particularly interesting addressing and reporting the factors associated not only with a vaccination-negative status, but also with women who did not complete the vaccination schedule. I warmly recommend to implement this analysis as it would improve the content of the paper.

3) please change 3D figure to a 2D one, as the 3D design impairs the proper appraisal of small differences

4) The comparison of Fig.2 and Fig.3 suggests that most of AUC from Fig.3 is associated with age, but age should be more properly assessed as being born before vs. after the most significant changes in the legal framework and formal recommendations towards HPV vaccine. Please fix it.

5) Please provide some information about the healthcare management of foreign-born individuals in Catalonia. I guess that EU/EEA foreigners should share the very same framework with Spain-born ones, while non-EU foreigners are expected to experience more difficulties in obtaining full medical support, including preventive interventions such as HPV.

6) Please explain how did you manage age in your analyses. As a continuous variable? as a multiple categorical one? as a dichotomous one? it is not clear.

Reviewer 2 Report

This manuscript leverages existing healthcare databases to evaluate factors associated with HPV vaccination status is a rural area of Spain. The most clear takeaway is that women under 30 were more likely to be vaccinated, which as authors point out, may be attributable to the timing vaccines were introduced in schools. However, there are several areas in presentation of data and of interpretation that must be clarified prior to publication. 

- Methods: Please provide justification for including only women aged 15-40 in the analysis when vaccination is available for younger children. Also please clarify how immigrant origin and ethnicity were defined, given the well-known in accuracies in health records regarding ethnicity. 

- Presentation of results: The results require significant streamlining and clarification. Currently, it is not consistent across tables whether the  percentage presented reflects the total in the x row or y column. Additionally, there a large number of tables that can be moved to supplemental material (table 4 and thereafter; figure 1) in favor of providing more detail regarding the results showing a 25-fold increase in vaccine completion. As framed now, it is not clear what the 25-fold is related to and as a compelling piece of data, would be served by more clarification.

- Discussion & overall framing: Currently, there are factors framed in this discussion as barriers or facilitators for vaccination. However, in an observational, epidemiologic study, those conclusions are overreaching. At most, an association can be identified but causality cannot. Additionally, many of the factors identified may be significantly impacted by factors not accounted for in this discussion, such as different care recommendations for screening of vaccinated versus unvaccinated persons or engagement in healthcare overall rather than a risk based behavior, unless there are timing elements for this analysis and the analyses regarding offspring and breastfeeding that were not detailed. Limitations need to include the limitations of the data sources selected. 

Other: The introduction is poorly structured to frame the question and variables included in this study. Additionally, the beginning of the results seems to be an overview of results more appropriate for the discussion than the results themselves which should focus on the findings and details supporting the findings, not a summary. 

The manuscript would benefit from editing for grammar and flow.

Reviewer 3 Report

Lara Colomé-Ceballos et al. investigated the factors associated with HPV vaccination in a rural area and identified the variables such as age, immigration origin, and HPV PCR test were influencing factors. Overall, the study was properly designed and clearly presented. I only have some minor comments.

The table layout is a little disorganized. The columns and rows can be switched so that the factors can be shown in one column. The values could be n (%) instead of two columns. Table 3 the comma is misused. The percentage of the cytology test in Table 4 is calculated differently from others.

Figure 1 could be moved after Table 1 in 3.1.

The vaccination rate in women without cervical cytology test is much higher than those with test. It’s regarded as a facilitator factor in line 333. Is there any discussion on this seemingly counterintuitive result?

Round 2

Reviewer 1 Report

Authors have diligently addressed all my concerns, therefore I'm endorsing the acceptance of this paper.

Author Response

The authors thank you for your work as a reviewer.